# Chromatographic Data in Statistical Analysis of BBB Permeability Indices

**DOI:** 10.3390/membranes13070623

**Published:** 2023-06-26

**Authors:** Karolina Wanat, Elżbieta Brzezińska

**Affiliations:** Department of Analytical Chemistry, Faculty of Pharmacy, Medical University of Lodz, 90-419 Lodz, Poland; elzbieta.brzezinska@umed.lodz.pl

**Keywords:** blood–brain barrier, BBB permeation, log BB, Kp,uu,brain, statistical modeling, data mining techniques, chromatographic retention data

## Abstract

Blood–brain barrier (BBB) permeability is an essential phenomena when considering the treatment of neurological disorders as well as in the case of central nervous system (CNS) adverse effects caused by peripherally acting drugs. The presented work contains statistical analyses and the correlation assessment of the analyzed group of active pharmaceutical ingredients (APIs) with their BBB-permeability data collected from the literature (such as computational log BB; Kp,uu,brain, and CNS+/− groups). A number of regression models were constructed in order to observe the connections between the APIs’ physicochemical properties in combination with their retention data from the chromatographic experiments (TLC and HPLC) and the indices of bioavailability in the CNS. Conducted analyses confirm that descriptors significant in BBB permeability modeling are hydrogen bond acceptors and donors, physiological charge, or energy of the lowest unoccupied molecular orbital. These molecular descriptors were the basis, along with the chromatographic data from the TLC in log BB regression analyses. Normal-phase TLC data showed a significant contribution to the creation of the log BB regression model using the multiple linear regression method. The model using them showed a good predictive value at the level of R^2^ = 0.87. Models for Kp,uu,brain resulted in lower statistics: R^2^ = 0.56 for the group of 23 APIs with the participation of k IAM.

## 1. Introduction

### 1.1. Prediction of CNS Availability

Predicting the possibility of drug penetration into the CNS is based on the behavior of drugs during pharmacokinetic distribution in the human body. This distribution is limited due to the existence of the blood–brain barrier (BBB), whose task is to maintain brain homeostasis by limiting the penetration of endogenous and exogenous chemical compounds. For this reason, the search for new, potential drugs should include their possible bioavailability in the CNS [1]. The blood–brain barrier is the most extensive “firewall”, which includes various defense mechanisms. In addition to the physical blockade, which is constituted by biological membranes connected by the tight and adherens junctions (TJs and AJs), there are also mechanisms for the active removal of xenobiotics (efflux transporters; P-glycoprotein) [2] or an enzymatic activity [3,4].

Biological experiments determining the level of drug distribution to the brain are the best source of information. However, these experiments are extremely time-consuming, expensive, and difficult to access in extensive screening studies, especially with the use of structure libraries. The most frequently used parameters in determining penetration through the BBB is log BB [5,6,7]. Log BB can be defined as the logarithm of the ratio of the drug concentration in the brain to the concentration of the drug in the blood in the state of dynamic equilibrium of the drug distribution between the blood and the brain, and it is considered as a convenient parameter in the assessment of BBB permeability [8]. Over the years, the Kp,uu,brain parameter has gained a lot of attention and its use is becoming more and more common in research [9,10,11]. In Table 1, the recent approaches in predicting BBB permeability in silico and in vitro are listed; computational methods seem to be more focused on the properties of the API, whereas in vitro modeling is based on the imitation of the BBB itself and the surrounding environment.

### 1.2. Computational Modelling of BBB Penetration

During preliminary research, drug design, and optimization, the simplest methods are used, often not even requiring the synthesis of the designed candidates. This is the phase of designing chemical structures, the construction of which allows one to achieve the expected physicochemical properties. Therefore, only in silico observation of their structure and the prediction of properties is possible.

Chromatography is an analytical technique for separating mixtures of substances; the mechanism of operation of this method is based on the differences in the affinity of the components of the mixtures toward the two phases present in the analytical system. The flow of the mobile phase through the stationary phase initiates the elution of compounds bound or adsorbed on the stationary phase. This process proceeds at a different rate for each component of the mixture. To describe this phenomenon in thin-layer chromatography (TLC), the retardation factor R_f_ is used, while in the case of high-performance column chromatography (HPLC), the retention factor values k are collected [22,23]. In our analyses, we tried to include the chromatographic retention data (TLC and HPLC) to increase the statistical modeling capabilities. Some authors have also attempted to implement chromatographic data to predict penetration into the CNS with satisfactory results [11], but the main focus is on HPLC, where the possibilities of the thin layer are investigated in this paper. Previous experiments [24,25,26] have shown a close relationship between the RP-18 TLC chromatographic data and bioavailability to the central nervous system. TLC is such a simple and economical option that it is worth updating its use from time to time, as it can be provided as a first step in chemical structure analysis before moving on to more expensive methods.

The experiment presented in the paper concerns the connection of retention data obtained from various chromatographic systems with the ability of drugs to penetrate into the central nervous system (log BB; CNS+/−). The aim of the work was to find a model of BBB penetration using physicochemical properties associated with retention data from the TLC and HPLC experiments. The dependent variables used in the modeling were two computational parameters labelled as B1 and B2, representing the log BB values. These parameters were adopted on the basis of the bibliography [24,25,26,27]. The calculation parameter B2 is quantitative and corresponds to log BB = 0.547 − 0.016 PSA [7,28]. The second parameter that describes log BB: B1 [29] was obtained from the online calculator. B2 and B1 are later described as indices of BBB permeability. Log BB was chosen as the main indicator of permeation through the BBB due to its popularity and the wide availability of its values for a large number of chemical compounds. The third dependent variable, Kp,uu,brain, was collected for only 29 APIs [30] out of all 181 drugs included in the study. An attempt was also made to build a simple regression model for this index.

The limits of the log BB index are different for many of the proposed BBB penetration prediction models. The optimal classification threshold is usually between 0 and −1 [27,31,32]. In one study, BBB+ (crossing the barrier) compounds were found to have a log BB value greater than 0 [33]. For BBB− (not crossing the barrier) compounds, a cut-off value of log BB <−0.3 was established [33]. The log BB limit = −0.52 is a logical division between BBB+ and BBB−. It corresponds to a 30% ratio of the concentration of the compound in the brain to the concentration in plasma [27]. Studies conducted earlier in the department defined the BBB+ limit as log BB ≥−0.9 [24,25,26]. Over the past 20 years, a number of different analyses have appeared to assess the bioavailability of drugs to the CNS using log BB values. The accuracy of these models ranges from 75% to 99% [34].

All mentioned BBB-permeability indices—B2, B1, and Kp,uu,brain—were taken into account when analyzing their relationship with the properties of the tested APIs. They are all correlated with the molecular descriptors at different levels, which is presented in the correlation matrix (Appendix A). Indicators B2 > −0.52 and B2 > −0.9 were also provided in the analysis where the API group is restricted to compounds with such log BB values only.

## 2. Materials and Methods

### 2.1. Chromatographic Experiments

A total of 181 active pharmaceutical ingredients were isolated from the pharmaceutical preparations. The purity of the isolated substances was checked by TLC chromatography and densitometric scanning. The obtained API was dissolved in 99.8% methanol to give 1 mg/mL solutions that were then used for TLC and HPLC chromatography.

APIs were subjected to chromatographic in column (HPLC) and thin layer (TLC) formats. TLC chromatography was carried out on 20 cm × 20 cm glass plates by Merck, covered with silica gel and the addition of a fluorescent indicator. Merck TLC silica gel 60 F254 was used for the normal phase (NP). In the reverse phase (RP), RP-2 silanized plates were used (Merck TLC Silica gel 60 RP-2 F254, silanized). The mobile phase was buffered with ammonium acetate to pH 7.4 (LACH-NER, ammonium acetate). The mobile phase composition was water:methanol:acetonitrile with the HPLC gradient grade (60:20:20 (*v*/*v*/*v*)).

Plates, after developing the chromatograms, were scanned using a densitometer (Desaga CD 60) with an analytical wavelength selected individually for drugs. The values of R_f_ (molecular descriptors: NP, RP) were determined.

Chromatographic data also came from an experiment with a HPLC IAM (immobilized artificial membrane) column (Regis Technologies Inc.: IAM.PC.DD.2, 10 µm; 4.6 mm × 10 mm). The column contains a standard membrane consisting of a phospholipid bilayer. Organic solvents (acetonitrile and methanol) and water (J.T. Baker HPLC) were used in HPLC. Sigma phosphate buffered saline tablets were used to prepare the phosphate buffer. The mobile phase was a mixture of 10 mM phosphate buffer at pH 7.4 and acetonitrile in the ratio of 80:20 (*v*/*v*) [35]. The mobile phase flow was set at 0.5 mL/min; detection was carried out with a UV–Vis detector. The obtained values of the retention factor (k IAM) were transformed into a logarithmic form and entered into the statistical analysis (descriptor: log k IAM).

### 2.2. Statistical Analyses and Molecular Descriptors (MDs)

Correlation analyses, analysis of variance (ANOVA), and multiple linear regression (MLR) were performed on a basic set of 181 active pharmaceutical ingredients (APIs). In recent years, more and more attention has been paid to data mining techniques that consist of searching for dependencies between variables in datasets. The set of the studied APIs was then analyzed using the MARSplines method, which is a non-parametric regression algorithm. The general mechanism of this method is described as multiple piecewise linear regression, built from basis functions. The boundaries of each section define the “applicability ranges” of the individual linear equations.

All statistics analyses were performed using STATISTICA 13.3 software (TIBCO Software Inc. Palo Alto, CA, United States) using the available functions. Molecular descriptors (MDs) connected to the physicochemical, pharmacokinetic, or chromatographic properties of APIs are listed in Table 2. The dataset for the 181 compounds and their structures are available in the Appendix A.

## 3. Results

### 3.1. Distribution of log BB and the Retention Values in the CNS Penetration Groups

The computational parameters B1 and B2 were examined first—these indicators were analyzed in terms of the distribution of their values in the CNS+/− groups (entering the CNS/not entering the CNS) [36] to check whether there was a significant difference in the log BB in both groups. The non-parametric Mann–Whitney U test remained due to the lack of normal distribution of B1 and B2 values in the CNS+/− groups (Table A1, Appendix B).

Among these indices, a better result was obtained with B2. Its median values in both the CNS+/− groups were statistically significantly different from each other, which was also clearly noticeable in the box plot (Figure 1). In a similar way, the distribution of the retention values from the chromatographic experiments (NP TLC, RP TLC, and HPLC IAM) was examined, together with the derivatives of these values by using the physicochemical properties—the NP and RP values were divided by the physicochemical properties of a given API. The obtained derivatives were as follows: NP/PSA; NP/PB; NP/log P; NP/Sa; NP/V; NP/MW; NP/log D. The same descriptors were calculated for the retention values from RP TLC (descriptor RP) and HPLC IAM (descriptor log k IAM) (Table A2).

The TLC descriptors showed several significant differences in the median values in the CNS+ and CNS− groups; these are all listed in Table A2 The best division visible in the boxplots was obtained for the descriptors RP, RP/PSA, and NP/PSA. The large influence of the polar surface area of the molecule PSA on the separation between the CNS+/− groups was shown when comparing the graphs for the retention factor, RP, and its derivative, RP/PSA (Figure 2).

Additionally, in the case of the HPLC descriptor log k IAM, a better division of the values between the CNS+/− groups was obtained, when the PSA parameter was also used (Table A3 and Table A4). When comparing both boxplots from Figure 3 directly, it should be noted that non-parametric tests were used for log k IAM due to the lack of normality of the distribution of this descriptor in the CNS+/− groups; the graph shows the medians and 25–75% of cases (Figure 3B). Normal distribution, however, was characterized as log k IAM/PSA values in both groups, and here the graph shows the means along with the standard deviation (SD) (Figure 3A).

The last analysis of this type was a simple ANOVA to check whether there were significant differences in the values of the TLC and HPLC descriptors in the groups of B2 > −0.9 and B2 > −0.52. A working code for the B2 values was adopted, where the limits for individual groups 1–2–3 are the mentioned −0.52 and −0.9:B2 code 1—below −0.9;B2 code 2—−0.89 to −0.52;B2 code 3—above −0.52.

The TLC descriptors that showed significant differences in the B2 code groups were NP and RP and their derivatives in combination with the percentage of plasma protein binding: NP/PB and RP/PB. The exception was that only the NP descriptor showed significant differences in the median values between group 1 and groups 2 and 3 (Table A5 and Figure 4). The rest of the parameters differed only between groups 1 and 3, that is, the extreme values of the B2 code influenced the differences in the retention values.

In the case of the IAM column, significant differences (also in the median values as non-parametric tests were used again) were revealed for the log k IAM and two derivatives: again in combination with PB: log k IAM/PB and additionally log k IAM/log P. These differences also only concerned the extreme groups B2 code: 1 and 3.

### 3.2. Regression Models with TLC Descriptors for B2 and B1 Indices

The next step of the study was the construction of regression models for the BBB permeation indices. For this purpose, descriptors from TLC, HPLC and their derivatives were used again in combination with their basic physicochemical properties (from Table 2). The aim of the MLR was to check whether chromatographic descriptors can be included among the factors influencing the level of log BB indices. Additionally, the usefulness of the physicochemical derivatives of the NP and RP descriptors was assessed in comparison with the standard values. Table 3 shows the aggregated MLR results for three dependent variables: B2, B2 > −0.52, and B2 > −0.9. Complete regression equations are available in Appendix B (Equations (A1)–(A10)).

The regression summary of the dependent variable B2 yielded the best result (R^2^ = 0.86 and R^2^ = 0.87); the other indices, B2 > −0.52 and B2 > −0.90, were less successful. They explained from 54 to 74% of the variance of the variability of these indicators. In the MLR model using all B2 values, the chromatographic data entered as one of the independent variables were both RP and NP. In models of a limited number of compounds (B2 > −0.9 and B2 > −0.52), it was also possible to introduce derivatives of the retention coefficient: RP/MW, RP/V, NP/MW, and NP/V. The highest score was obtained for the NP descriptor. The resulting mathematical model explained 87% of the variance of variable B2 (Equation (1)) (Figure 5).
B2 = 0.652 − 0.167 HA + D − 0.215 NP + 0.007 eL + 0.037 PhChargR = 0.9332; R^2^ = 0.8713; R^2^ adj. = 0.8624; F(5, 158) = 240.15; *p* > 0.000; s = 0.24; N = 151(1)

In all multiple linear regression equations of the analyzed indices, B2, B2 > −0.52, and B2 > −0.90, there were MDs related to the number of possible hydrogen bonds formed by the tested compounds: HA—acceptors and HA + HD—sum of acceptors and donors. The second important parameter that appeared in this equation was the charge of the molecule in physiological conditions (i.e., at pH around 7.4—PhCharg and the energies of the lowest unoccupied molecular orbital (eL) or the difference between HOMO and LUMO (eH-eL)). For B2 > −0.52 and B2 > −0.90, other descriptors were also common: Sa (surface area of the molecule) and PB (level of plasma protein binding). These physicochemical parameters, related to the “rule of five” described above, confirm their relationship with the description of CNS bioavailability.

The MLR was performed using another computational parameter that describes log BB: B1, considered as the alternative to B2. The direct correlation between B2 and B1 was R = 0.52 for 143 cases of APIs; for B2 > −0.9 and B2 > −0.52, such correlation was lower, around 0.34–0.38 (Table 4).

The only model for the B1 indicator that included TLC descriptors was successfully built with the derivative of the RP retention value, namely, RP/MW. The model explained only 53% of the B1 variance, and again, the number of hydrogen bond donors and acceptors ranked in first place among the factors determining B1 variability. Although the influence of the RP/MW descriptor stood out from other independent variables in the model, it is also together with HA + D, as the only one qualified as statistically significant, with a significance level of *p* > 0.05 (Equation (2)) (Figure 6).
B1 = 0.630 − 0.189 HA + D + 0.109 V − 2.091 RP/MW − 0.040 pKa + 0.021 eL + 0.125 SaR = 0.7323; R^2^ = 0.5314; R^2^ adj. = 0.5087; F(1, 113) = 21.61; *p* > 0.000; s = 0.69; N = 134(2)

### 3.3. Regression Models with HPLC IAM Descriptors for B2, B1 Kp,uu,brain Indices

The HPLC IAM descriptor was also included in the regression equations of the B2, B2 > −0.52, and B2 > −0.90 indices with less success, so it was possible to introduce its derivatives; only log k IAM/PB participated in building the model for B2 > −0.52. The best result was obtained for B2; the coefficient of determination was 0.89, which was the highest of all MLR models (Table 5).
B2 = 0.598 − 0.171 HA + D + 0.013 eL + 0.035 log k IAM + 0.016 log U/D − 0.0007 MWR = 0.9513; R^2^ = 0.8934; R^2^ correct. = 0.8924; F(5, 118) = 197.87; *p* > 0.000; s = 0.19;N = 124(3)

Unfortunately, the log k IAM parameter entered the model (Equation (3)) with a rather high level of significance, *p* = 0.142, which may not fully prove its usefulness in predicting the B2 indicator. A similar situation occurred with the regression model for the B1 indicator (Equation (4)); here too, the log k IAM parameter entered the equation, with a *p* value above the assumed significance level of 0.05 (*p* = 0.093). The statistics of the model were also lower than those of the B2 index; R^2^ was only 0.51.
B1 = −0.030 − 0.172 HA + D + 0.271 V + 0.031 eL + 0.004 MW + 0.159 log k IAM − 0.027 pKaR = 0.7112; R^2^ = 0.5143; R^2^ correct. = 0.4842; F(6, 117) = 20.03; *p* > 0.000; s = 0.70;N = 124(4)

On the other hand, the IAM retention factor, descriptor k IAM, was the only chromatographic data that entered into the MLR model for the dependent variable Kp,uu,brain. The equation was built on a small group of compounds because only 23 of them were involved in the model. The coefficient of determination was close to the result from the B1 model, R^2^ = 0.56, so only slightly more than 50% of the variation in Kp,uu,brain can be explained by a model (Equation (5)) (Figure 7).
Kp,uu,brain = 3.541 + 0.071 k IAM − 0.541 log D − 0.331 DM − 0.836 PhChargR = 0.7542; R^2^ = 0.5641; R^2^ correct. = 0.4681; F(4, 18) = 5.820; *p* > 0.0291; s = 1.10; N = 23(5)

### 3.4. Data Mining Models with Chromatographic Data

The MARSplines method was carried out to check whether the results of the MLR models could be improved by using a different regression algorithm. Descriptors used in model building were the same as those that occur in MLR models; dependent variables were again the B2, B1, and Kp,uu,brain indices.

The statistics of the B2 equations with the descriptors NP and RP were slightly lower than those obtained from the MLR; the coefficients of determination were 0.85 and 0.82, respectively (compared to 0.87 and 0.86 from multiple linear regression). The equation with the log k IAM descriptor had the same R^2^ value as in the MLR model (0.89), but again, this chromatographic descriptor did not reach a significance level of *p* below 0.05 (Table 6).

For parameter B1, the regression result was also lower than that for the MLR (0.46 compared to 0.53) (Figure 8).

An attempt was made to perform a MARSplines analysis for the variable Kp,uu,brain. The results were, however, even lower than in the MLR model; the R^2^ was 0.37 in the best descriptor configuration, the retention factor k IAM was re-entered into the model, and was actually the only factor significantly affecting the modeling of the Kp,uu,brain values in this small group of analyzed compounds (N = 29).

## 4. Discussion

Comparison of the median values (Mann–Whitney U test) of the computational indices B1 and B2 reflecting the log BB in the CNS+/− penetration groups showed that only the B2 values significantly differed between the ‘+’ and ‘−’ groups. Therefore, more attention was paid to the B2 indicator in further proceedings and regression modeling. Similar analyses concerning the distribution of values in the CNS+/− groups were performed for the chromatographic descriptors NP, RP, and log k IAM and their derivatives, combined with their physicochemical properties. The descriptors alone showed significant differences in the medians in the CNS+/− groups. In the case of their derivatives, the physicochemical PSA parameter revealed a large influence because the chromatographic descriptors with its participation (RP/PSA; NP/PSA and log k/PSA) were characterized by the greatest difference between the medians (in the case of log k IAM/PSA − means) in the CNS+ and CNS− groups, which can be seen in the boxplots (Figure 2 and Figure 3)

Regression analyses carried out in the group of 181 APIs confirmed that physicochemical properties such as hydrogen binding capacity (HA, HD or HA + HD), the level of physiological charge (PhCharg), and the energy of the lowest unoccupied molecular orbital (eL) were the most important descriptors in regression models for the B2 index. They appeared most often in the resulting equations and usually entered them with a level of *p* < 0.05.

Statistical analysis of the chromatographic data (descriptors NP, RP, and log k IAM) proved the usefulness of these retention coefficients to predict the log BB values. The TLC data entered the MLR models with good results, and the best statistics were achieved for the NP descriptor. The model with this participation explained 87% of the variance of the B2 indicator, and the descriptor itself significantly contributed to the regression modeling (*p* = 0.012). The RP descriptor turned out to be less successful: for B2, it was not possible to enter it into the model with *p* < 0.05 (Equations (A1) and (A2) in Appendix B). The situation improved for limited groups of compounds (i.e., the B2 > −0.52 index (N = 89), where RP descriptor managed to be included in the equation with a level of *p* = 0.0409).

NP derivatives combined with physicochemical parameters were also used in MLR models: the NP/MW parameter was applied in the model for the B2 and B2 > −0.52 indices, while NP/V appeared in the equation for B2 > −0.9. In all cases, these descriptors entered the equations with *p* < 0.05 (i.e., their contribution to the modeling of the regression models was significant). RP/MW and RP/V also appeared in the MLR model as the only derivatives of the TLC data. This shows the significant influence of molar mass (MW) and molecular volume (V) in combination with the retention data on log BB modeling. These physicochemical properties (MW and V) alone did not appear often in the equations.

The regression analysis for the B1 index was less successful; with the TLC data, only one model could be built that explained 53% of the B1 variance. The chromatographic descriptor that contributed to this model was RP/MW (*p* = 0.026).

Retention coefficients obtained from high-performance liquid chromatography with a column containing immobilized artificial membrane (log k IAM), despite having quite promising primary correlations with the log BB indices (R = 0.39 for B1 and B2; Table A6) and had a smaller share in regression models (for B2 and B1), their *p* values did not usually exceed the assumed significance level of 0.05. However, the retention coefficient k IAM was the only one to be introduced into the MLR equation for the dependent variable Kp,uu,brain. The model was built for a small group of APIs (N = 23), but k IAM significantly contributed to the development of this model (*p* = 0.0002), along with the coefficient of distribution, log D (*p* = 0.023).

The regression method (MARSplines) in the field of data mining was applied to check whether better model statistics could be obtained by using a different regression algorithm. The results obtained using it were comparable or, unfortunately, lower than those obtained using the simplest multiple regression. The highest R^2^ = 0.85 in the MARSplines model was again obtained for the NP descriptor; RP resulted in R^2^ = 0.82; both parameters were statistically significant. Log k IAM did not reach the level of *p* < 0.05 as in the MLR model, although the equation was characterized by a very high coefficient of determination.

The B1 indicator also scored lower in the data mining algorithm; R^2^ dropped from 0.53 (MLR) to 0.46 (MARSplines) with the RP/MW descriptor.

Attempts to model the Kp,uu,brain indicator resulted in the creation of models with R^2^ = 0.56 (MLR) and 0.37 (MARSplines), which were created using the retention coefficient k IAM, distribution coefficient: log D, molecule dipole moment: DM, and charge in pH 7.4: PhCharg. The chromatographic descriptor had a significant impact on the creation of both models, however, the small size of the API group with the collected Kp,uu,brain data (N = 29) does not allow for further conclusions. The high correlation with the RP index also seems promising (R^2^ = 0.51, Table A6). Further analyses of the Kp,uu,brain indicator are planned, with more APIs participating in the study.

## 5. Conclusions

Retention data from the TLC experiments in combination with simple molecular descriptors (number or hydrogen acceptors and donors: HA and HD, energy of the lowest unoccupied molecular orbital: eL, physiological charge: PhCharg, molecular mass: MW etc.) allowed us to build several models of the prediction of log BB. The highest results were obtained with the NP TLC data (retardation coefficient R_f_ from the normal-phase TLC plate) This chromatographic descriptor is applicable in this type of statistical modeling and carries valuable information on the pharmacokinetic properties of a group of various APIs. The HPLC data seemed to show some influence on the Kp,uu,brain index, however, these analyses should be repeated on a larger group of APIs to see if this correlation persists.

Derivatives of the NP descriptor, combined with the molar mass and volume of the molecule, NP/MW and NP/V, respectively, were also used in the log BB regression equations with good results. Polar surface area derivatives (NP/PSA and RP/PSA) showed the best distribution of values between the CNS+ and CNS− groups. Based on this information, it can be concluded that the derivatives of chromatographic data may increase the usefulness of retention values (R_f_ or log k) in the statistical analyses of BBB permeation.

## Figures and Tables

**Figure 1 membranes-13-00623-f001:**
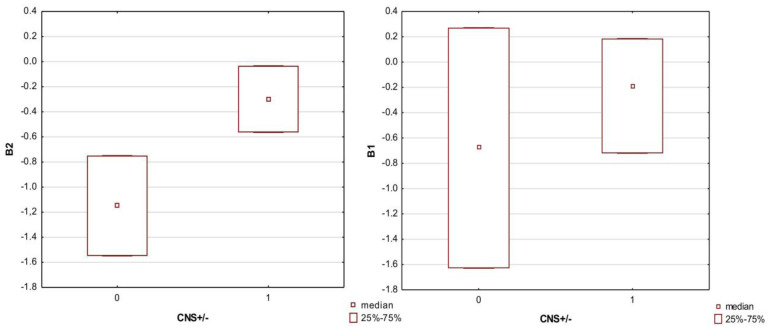
Boxplot of the B1 and B2 indices in the CNS+ and CNS− groups. Medians were compared due to the application of non-parametric tests. Code 0—CNS− group; code 1—CNS+ group.

**Figure 2 membranes-13-00623-f002:**
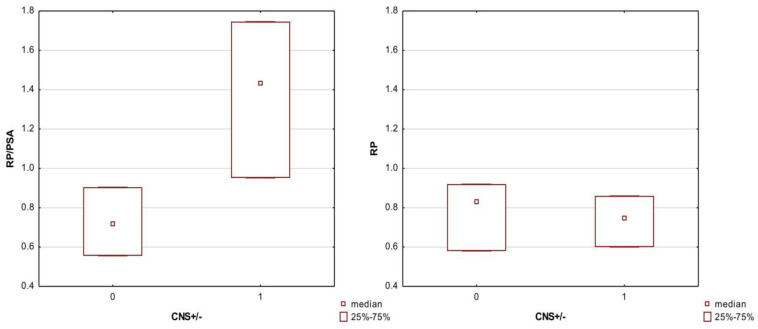
Boxplot of the RP and RP/PSA descriptors in the CNS+ and CNS− groups. Medians were compared because non-parametric tests were used. Code 0—CNS− group; code 1—CNS+ group.

**Figure 3 membranes-13-00623-f003:**
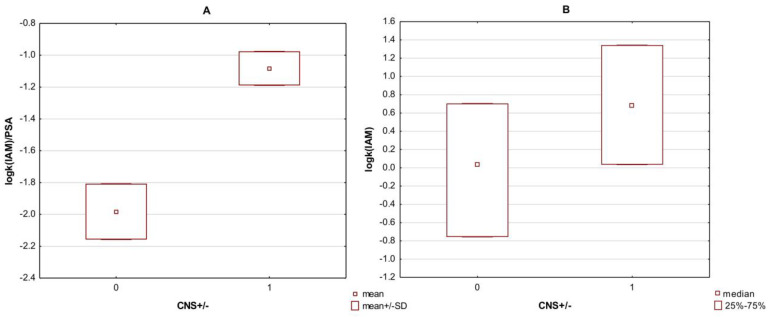
Boxplot of the log k IAM (**B**) and log k IAM/PSA (**A**) values in the CNS+ and CNS− groups. Medians were compared in the case of log k IAM. Mean values were compared for log k IAM/PSA. Code 0—CNS− group; code 1—CNS+ group.

**Figure 4 membranes-13-00623-f004:**
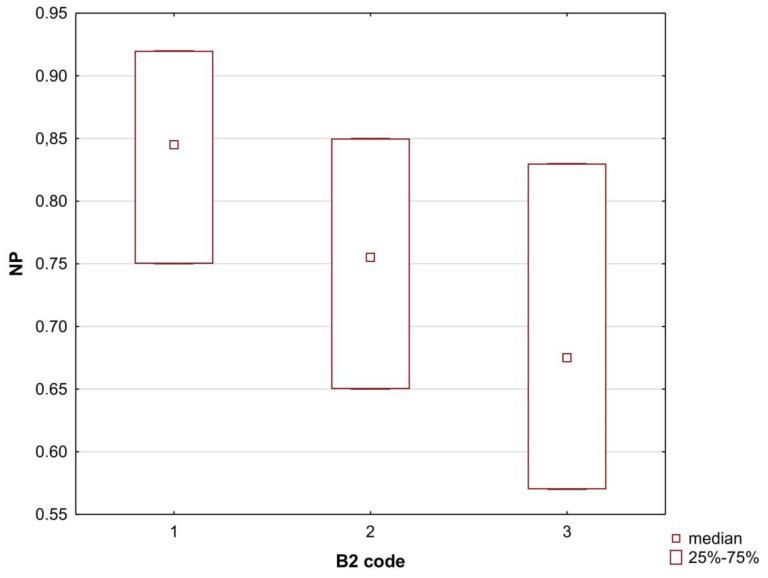
Boxplot of the NP median values in the B2 code groups: 1, 2. and 3. According to the Kruskal–Wallis ANOVA, the differences between medians in groups 1 and 2 and groups 1 and 3 were statistically significant; *p* < 0.05.

**Figure 5 membranes-13-00623-f005:**
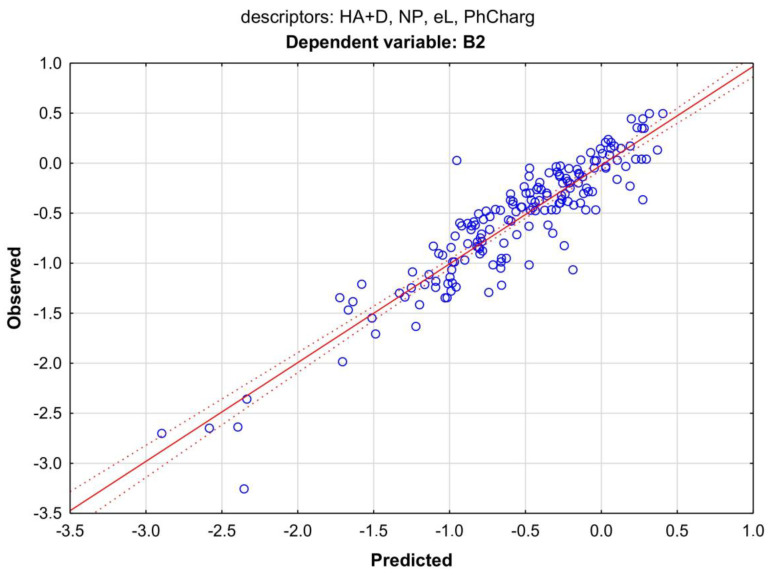
Scatter plot of the MLR model from Equation (1). Dependent variable: B2; predictors: HA + HD, NP, eL, and PhCharg. Coefficient of determination R^2^ = 0.87.

**Figure 6 membranes-13-00623-f006:**
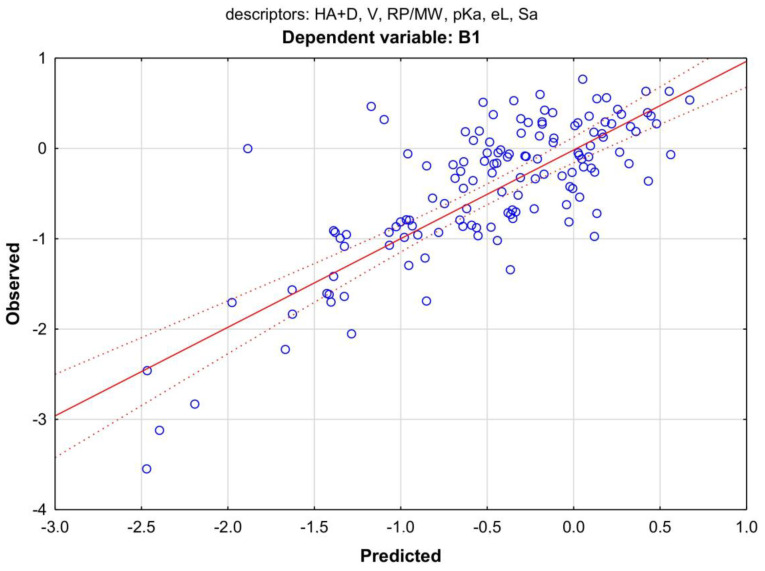
Scatter plot of the MLR model from Equation (1). Dependent variable: B1; predictors: HA + HD, V, RP/MW, pKa, eL, and Sa. Coefficient of determination R^2^ = 0.53.

**Figure 7 membranes-13-00623-f007:**
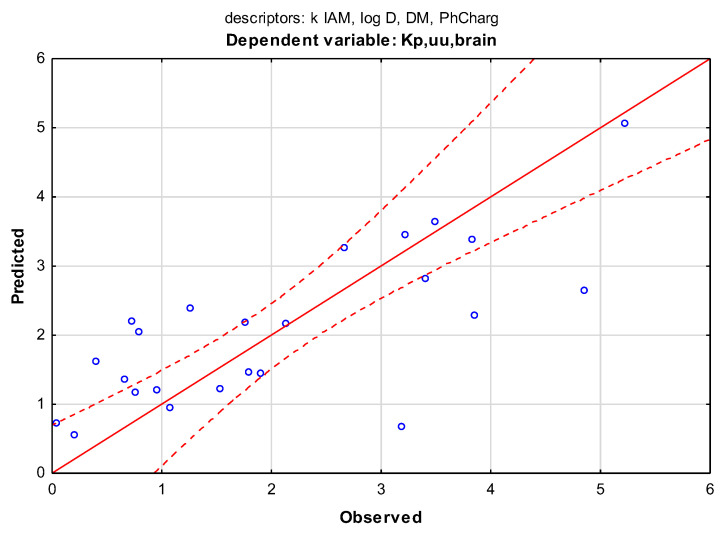
Scatter plot of the MLR model from Equation (5). Dependent variable: Kp,uu,brain; predictors: k IAM, log D, DM, and PhCharg. Coefficient of determination R^2^ = 0.56.

**Figure 8 membranes-13-00623-f008:**
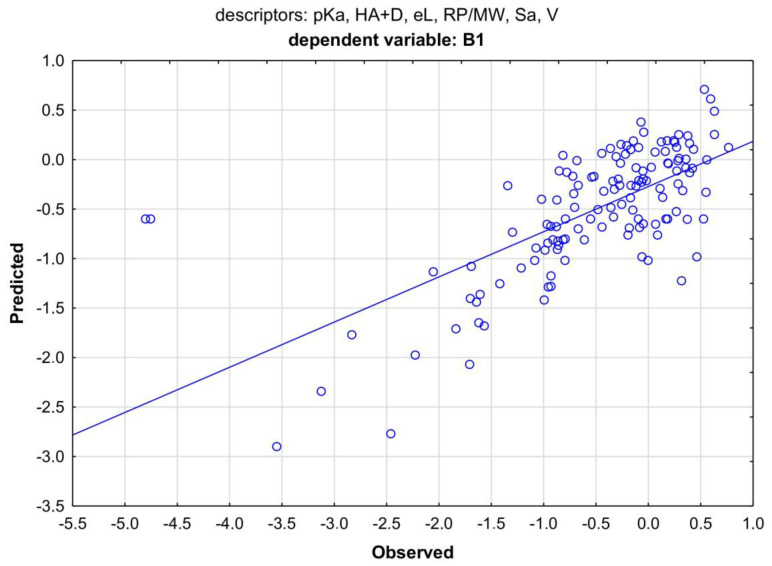
The MARSplines model for B1. Two basis functions were used in the model algorithm. Coefficient of determination R^2^ = 0.46.

**Table 1 membranes-13-00623-t001:** In silico and in vitro approaches in predicting BBB penetration. In silico methods are described using molecular descriptors used in model building; in vitro methods by the type of BBB model used.

Prediction of BBB Penetration	
In Silico	In Vitro
Solvation free energies in various solvents of different polarities [12]	Single endothelial cell monolayer model [13]
1D, 2D, and 3D molecular descriptors and fingerprints of a molecule [14]	Stem cell modeling of the BBB: neural progenitor cells (NPCs), the precursors to neurons, astrocytes or oligodendrocytes [15]
Molecular descriptors: carboxylic acid group, polar surface area (PSA)/hydrogen-bonding ability, lipophilicity, and molecular charge [16]	Co-culture model (brain microvascular endothelial cells: BMECs with astrocytes or pericytes) [17]
1D and 2D physicochemical properties, molecular access system fingerprints -MACCS and substructure fingerprints [18]	The microfluidic BBB [13]—microfluidic devices to mimic biological environments: -observation of specific markers of TJs [19] -measuring TEER [20] -permeability assessment [21]
HPLC data (IAM, HSA and AGP columns) and molecular descriptors to model brain disposition of drugs: K_p,uu,brain_ [11]	

**Table 2 membranes-13-00623-t002:** List of molecular descriptors (MDs) used in the statistical analyses.

MD	Description	Source
B2	Computational parameter, determines penetration through the blood–brain barrier: log BB = 0.547 − 0.016 PSA	[7,28]
B1	Computational parameter, corresponds to log BB	SwissADME
CNS+/−	Describes the bioavailability in the CNS	Drugbank
eL	Energy of the lowest unoccupied molecular orbital	Hyperchem
eH	Energy of the highest occupied molecular orbital	Hyperchem
eL-eH	Ionization capacity	Hyperchem
HA	Hydrogen bond acceptors	Hyperchem
HD	Hydrogen bond donors	Hyperchem
Kp,uu,brain	Unbound brain-to-plasma drug partition coefficient	[30]
log D	Distribution coefficient	ACD Labs
log k IAM	Logarithm of retention factor from HPLC IAM	HPLC
log P	Partition coefficient	Hyperchem
log U/D	Describes the extent of ionization, calculated from pKa, according to the equations: pKa—pH (acids) or pH—pKa (bases)	ACD Labs
MW	Molecular weight	Hyperchem
NP	The R_f_ from NP TLC plate	TLC
PB	Protein binding	Drugbank
PhCharg	Charge of a compound in physiological environment	Drugbank
PSA	Polar surface area of a molecule	Hyperchem
RP	The R_f_ from RP-2 TLC plate	TLC
Sa	Surface area of a molecule	Hyperchem
V	Molecular volume	Hyperchem

**Table 3 membranes-13-00623-t003:** The MLR results of the B2, B2 > −0.52, and B2 > −0.9 indices for the NP and RP TLC retention data.

	B2	B2 > −0.52	B2 > −0.9
RP	R = 0.9214 R^2^ = 0.8643 R^2^ adj. = 0.8589 F(5, 158) = 200.15; *p* > 0.000 N = 164	R = 0.7374 R^2^ = 0.5433 R^2^ adj. = 0.5091 F(6, 82) = 16.18; *p* > 0.000; N = 89	R = 0.8432 R^2^ = 0.7112 R^2^ adj. = 0.6924 F(6, 104) = 42.27; *p* > 0.000 N = 111
RP derivatives	RP derivatives do not enter the model	R = 0.7287 R^2^ = 0.5422 R^2^ adj. = 0.5133 F(4, 84) = 24.24; *p* > 0.000 N = 89 derivative: **RP/MW**	R = 0.7842 R^2^ = 0.6011 R^2^ adj. = 0.5924 F(4, 106) = 40.30; *p* > 0.000 N = 111 derivative: **RP/V**
NP	R = 0.9332 R^2^ = 0.8713 R^2^ adj. = 0.8591 F(5, 158) = 240.15; *p* > 0.000 N = 151	NP does not enter the model	R = 0.8572 R^2^ = 0.7412 R^2^ adj. = 0.7156 F(7, 103) = 41.43; *p* > 0.000 N = 111
NP derivatives	R = 0.9322 R^2^ = 0.8734 R^2^ adj. = 0.8555 F(6, 144) = 160.07; *p* > 0.000 N = 151 derivative: NP/MW	R = 0.7813 R^2^ = 0.6122 R^2^ adj. = 0.5843 F(7, 81) = 18.249; *p* > 0.000 N = 89 derivative: NP/MW	R = 0.8142 R^2^ = 0.6481 R^2^ adj. = 0.6343 F(5, 105) = 38.703; *p* > 0.000 N = 111 derivative: NP/V

**Table 4 membranes-13-00623-t004:** Correlations, N = 143 (missing data were removed by cases). All correlations were statistically significant, with *p* < 0.05.

N = 143	Mean	SD	B1	B2	B2 > −0.9	B2 > −0.52
B1	−0.4901	0.9472	1.0000	0.5192	0.3776	0.3350
B2	−0.6152	0.6802	0.5192	1.0000	0.7832	0.7491
B2 > −0.9	0.7273	0.4469	0.3776	0.7832	1.0000	0.6708
B2 > −0.52	0.5455	0.4997	0.3350	0.7491	0.6708	1.0000

**Table 5 membranes-13-00623-t005:** The MLR results of the B2, B2 > −0.52, and B2 > −0.9 indices for the HPLC IAM retention data.

	B2	B2 > −0.52	B2 > −0.9
log k IAM	R = 0.9521 R^2^ = 0.8938 R^2^ correct. = 0.8877 F(5, 118) = 197.87; *p* < 0.000 N = 124	R = 0.8687 R^2^ = 0.7644 R^2^ correct. = 0.7456 F(5, 60) = 37.67; *p* < 0.000 N = 66	R = 0.8933 R^2^ = 0.8024 R^2^ correct. = 0.7714 F(9, 73) = 31.85; *p* < 0.000 N = 83
log k IAM derivatives	log k IAM derivatives do not enter the model	R = 0.8742 R^2^ = 0.7633 R^2^ correct. = 0.7434 F(5, 60) = 37.15; *p* < 0.000 N = 66 derivative: log k IAM/PB	log k IAM derivatives do not enter the model

**Table 6 membranes-13-00623-t006:** Comparison of the MLR and MARSplines models for the B2 index using the NP, RP, and log k IAM descriptors.

	NP	RP	log k IAM
MLR	R^2^ = 0.8743 R^2^ correct. = 0.8633 N = 151	R^2^ = 0.8574 R^2^ correct. = 0.8642 N = 164	R^2^ = 0.8912 R^2^ correct. = 0.8895 N = 124
MARSplines	R^2^ = 0.8533 R^2^ correct. = 0.8524 N = 169	R^2^ = 0.8218 R^2^ correct. = 0.8242 N = 169	R^2^ = 0.8944 R^2^ correct. = 0.8891 N = 116
Descriptors in the models	HA + D, NP, eL, PhCharg	HA + D, eL, RP, log P, PhCharg	HA + D, eL, log U/D, log k IAM, MW

## Data Availability

Not applicable.

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
