# Peer review of "Chromatographic Data in Statistical Analysis of BBB Permeability Indices"

_membranes, 2023, doi:10.3390/membranes13070623_

Round 1

Reviewer 1 Report

S2#. This work comprises several different steps…
I suggest authors to emphasis details in each case to evaluate this binary classification problem in therms of “confusion matrix” (for initial guidance… https://en.wikipedia.org/wiki/Confusion_matrix) Some examples: In section “3.1 Distribution of Log(BB) and retention values in CNS penetration groups” > what was the real accuracy in this binary classification?
> what was sensitivity (TPR), specificity (TNR), etc… ---- S3#. Tables 4 and 5 > please include in table footer with explanations for “U”, “Z” and “p”... ---- S4#. Significant figures (starting from section 3.2…) > accordingly to “Uncertainty estimation and representation” standard conventions, uncertainties (like SDEP) should be represented with a maximum of 2 significant figures > coefficient of determination (R^2; R^2_adj) – should be represented wit at least 3 significant figures (usually seen with 4 decimals...) ---- S5#. Figures 6 to 8 > since predicted value is estimated and less accurate then the actual value (experimental), it is more correct to represent ESTIMATED (in the vertical axis, as dependent variable) and REAL OBSERVED value (in the horizontal axis, as independent variable) > in these “recovery plots”, to better check the accuracy, its preferable to use isoscaled axis and draw the bisector of first quadrant (45º slope); then overlay your data points ---- S6#. In Table A1 > add vertical separator lines between columns 1-2 and 3-4 ---------------------------------- 3. Questions: Q1# Figures 1 to 4: > these box-plots only represents 50% of actual considered data (Q1-Q3 interval)? > How many samples were actually correctly classified (CNS+/-) between used datasets? ---- Q2# Lines.# 181-182 “The best division… and NP/PSA" > please check again because I suspect that with “NP” you may have a similar result? > maybe… all results with p < 0,010 will show a very good separation? ---- Q3#. In “3.2 Regression models with TLC...” and “3.3 Regression models with HPLC...” > What was the criteria for “model choice” (parameter optimization)? ---- Q4#. In equations (A1) to (A10) > R^2_corr = R^2_adjust ? > the meaning of SDEP? (error of prediction? What kind of Cross Validation was performed?) > dividing SDEP by mean response value → you will get a relative SDEP which is more elucidative of expected error...

Reviewer 2 Report

The reviewer carefully read out the manuscript and gave some comments as follows. The authors used many substrates to elucidate the corelation between BB permeation and HPLC retention behavior. The reviewer understood the importance of this study. Meanwhile, it was supposed that the flow of figures that tells us the story line of the study was unlikely to be rich,  e.g. a display of tables for dscriptors and some figures to compare the correlation coefficients. That is to say, the present version of manuscript let the reviewer strolgly bored. If such a point is improved, the readability (plainfullness) would be also improved. Again, reconsider the improvement of this manuscript to avoind the difficulty in understanding of contents, if you would?

(1) Abstract: Some abberviations were used without  their diffinition, e.g. CNS, MD, MRL.

(2) Introduction: Authors should describe what is the academic problem to be focused on in this article.

(3) Materials and Methods: What kinds of atifical membranes were used in the column as stated in "Chromatographic data ... HPLC IAM (immo- 125 bilized artificial membrane) column ..." (lines 125-127)?

(4) In Figure 3, the title of y-axis, "logIAM/PSA", should be revised as "logkIAM/PSA", shouldn't it?  

(5) Show the definition of logBB that is a kenel of this article.

(6) Table 2: eH: "ebergy" -> "energy"?

(7) The reviewer could not understand how the result of box-plots in Figs.1-4 could be logically connected to the correlation between two parameters (Figs.5-8).

(8) Line 402: "R = 0.51" -> "R2 = 0.51"
